# Dairy Food Consumption Is Associated with Reduced Risk of Heart Disease Mortality, but Not All-Cause and Cancer Mortality in US Adults

**DOI:** 10.3390/nu15020394

**Published:** 2023-01-12

**Authors:** Yanni Papanikolaou, Victor L. Fulgoni

**Affiliations:** 1Nutritional Strategies, 59 Marriott Place, Paris, ON N3L 0A3, Canada; 2Nutrition Impact, 9725 D Drive North, Battle Creek, MI 49014, USA

**Keywords:** dairy foods, usual intake, NHANES, mortality, cancer, cardiovascular disease

## Abstract

Previous evidence has linked animal protein intake, including dairy foods, with an increased risk in mortality from all-causes and certain chronic diseases, including cancer and heart disease. The objective of the current analysis was to examine associations between total dairy consumption with mortality from all-causes, cancer, and heart disease. Data for adults (≥19 y; *n* = 54,830) from the *Third National Health and Nutrition Examination Survey* (NHANES) and NHANES 1999–2014 were linked with mortality data through 2015. Individual usual intake for dairy foods were estimated using the National Cancer Institute method. Hazard ratio (HR) models were fit for mortality types (all cause, cancer, heart disease) and measures of usual intakes of dairy. Multivariable analysis further adjusted for age, gender, ethnicity, waist circumference, smoking status, education level, chronic condition status (i.e., based on cancer, myocardial infarct, and diabetes/diabetes medication reported), weight loss attempts, and % kcal from animal protein. No associations were seen between dairy food intake and mortality risk from all-causes [HR = 0.97; confidence intervals (CI): 0.85–1.11; *p* = 0.67], and cancer [HR = 0.95; CI: 0.75–1.20; *p* = 0.65] when comparing the lowest quartile to the highest quartile of consumption. Dairy food consumption was associated with a 26% reduced risk for heart disease mortality when comparing the lowest quartile to the highest quartile [HR = 0.74; CI: 0.54–1.01; *p* = 0.05]. Further analyses in different age groups showed that dairy food consumption was associated with 39% and 31% reduced risk for heart disease mortality in older adults 51–70 and ≥51 y, respectively [adults 51–70 y: HR = 0.61; CI: 0.41–0.91; *p* = 0.01; adults ≥51 y: HR = 0.69; CI: 0.54–0.89; *p* = 0.004]. These results contradict previous findings that have linked dairy foods to increased mortality risk. Further, dairy foods as part of a healthy dietary pattern, may help lower heart disease mortality risk.

## 1. Introduction

Higher dietary intakes of animal protein have been associated with an elevated mortality risk from all-cause, cardiovascular disease (CVD), and cancer. Dairy protein remains a predominant animal protein source in current US dietary patterns, with estimates showing all milk types (i.e., whole, non-fat, reduced fat and low-fat milk) combined contribute 28% of total dairy protein intake in a typical US adult dietary pattern [1]. Furthermore, a substantial level of evidence, stemming from both randomized clinical trials and epidemiological studies, supports dairy foods as part of healthy dietary patterns [2]. Indeed, both the previous two iterations of the Dietary Guidelines for Americans have recommended low-fat and fat-free dairy foods as part of healthy dietary patterns [3,4]. *The 2020–2025 Dietary Guidelines for Americans* (2020–2025 DGA) emphasizes the increased consumption of nutrient-dense foods and beverages, including low-fat and fat-free dairy products, due to their vitamin, mineral and health promoting components. Further, the 2020–2025 DGA policy document is supported by consistent and accumulating evidence showing positive associations between dietary patterns and health outcomes [4]. Specifically, strong or moderate evidence reviewed by the 2020 Dietary Guidelines Advisory Committee (2020 DGAC) concluded healthy dietary patterns, including patterns that include low-fat and fat-free dairy foods, are associated with lower risk of all-cause mortality, lower risk of cardiovascular mortality, reduced risk of type 2 diabetes, and certain cancers [4].

In contrast to the scientific position adopted by the 2020 DGAC, previous research using cross-sectional data reported positive associations between animal protein consumption and health outcomes. Data in adults using NHANES III found that higher protein intake was associated with a 75% increase in overall mortality and a 4-fold increased risk of cancer and diabetes in older adults. These findings became null or attenuated if the protein source consumed was derived from plants [5]. In contrast, higher protein intake in adults over 65 years of age was associated with reduced risk of cancer-related and all-cause mortality [5] despite that age is generally considered a primary risk factor for cancer mortality [6]. Nonetheless, the investigators did not consider usual protein intakes (i.e., an established measure of long-term intake), as recommended by the National Cancer Institute methodology when assessing mortality risk [7], suggesting substantial methodological concerns, as usual intake has been established as a validated measure of habitual intake and a critical dietary intake assessment method when examining mortality risk outcomes.

The objective of the current study was to examine associations between total dairy usual intake with mortality risk from all-causes, cancer, and heart disease using data from the *Third National Health and Nutrition Examination Survey (NHANES)* and NHANES 1999–2014 with links to mortality data through 2015. It was hypothesized that usual protein intakes from dairy foods would be associated with a decreased risk for mortality from all-causes, as well as CVD and cancer, but that this relationship may be dependent on age.

## 2. Experimental Section

The US NHANES is a database conducted by the National Center for Health Statistics (NCHS) of the Center for Disease Control and Prevention (CDC), and represents a nationally representative, cross-sectional survey of U.S. free-living, civilian residents. Thorough data collection methods and analytics of NHANES have been previously documented in peer-reviewed publications [1,8,9] and are available online [10]. Comprehensive ethical considerations, including, but not limited to, informed consent and privacy protections for all study subjects have been undertaken, and the NHANES protocol has been previously approved by the Research Ethics Review Board at NCHS. For the current analysis, data were obtained from NHANES III, which comprised data collected from 1988 to 1994 and NHANES 1999–2014, and included energy, protein, and total dairy intake data on all adults ≥19 years old. The analysis contained 15,937 participants following exclusions for unreliable intake data, mortality follow-up ineligible data, and pregnant or lactating females (refer to Figure 1 for detailed exclusion values). Energy and protein intake data from NHANES III dietary intake files and energy and protein intakes for NHANES 1999–2014 are from the relevant United States Department of Agriculture (USDA) Food and Nutrient Database for Dietary Studies (FNDDS) [11]. FNDDS are databases that provide the nutrient values for foods and beverages reported in What We Eat in America (WWEIA), the dietary intake component of NHANES for each data release [12]. Total dietary intake was obtained from MyPyramid Equivalents database (MPED) for NHANES III and from relevant MPED and Food Patterns Equivalent Databases (FPED) for NHANES 1999–2014 [13,14]. Animal protein was determined by a scientific approach previously published [15], which defined animal protein foods using MPED/FPED variables for type of protein food. 

### Methods, Subjects and Statistical Analysis

SAS 9.4 (SAS Institute, Cary, NC, USA) was used for all analyses and NHANES survey parameters including examination weights, strata and primary sampling units were implemented. Data in men (≥19 years old, *n* = 27,200) and women (*n* = 27,630) were linked with mortality data through 2015 from the *National Death Index*—a centralized registry of all US deaths [16]. Individual usual intakes were estimated for energy, animal protein (as % kcal) and dairy intake variables (using the National Cancer Institute method (Version 2.1), which has been previously described [7]. Covariates for usual intake estimation were dietary recall sequence, Dietary Reference Intake age group, weekend (Friday, Saturday or Sunday) intakes, and race/ethnicity. Hazard ratios (HR) models were fit for mortality types (all-cause, cancer, CVD) with usual dairy food intake. Quartiles of total dairy food intake were established.

Survival models were created using follow-up months from examination date reported in NHANES. While usual dairy intake was the primary variable of interest (with the lowest quartile of usual dairy intake being the reference group (HR + 1.0), other variables in the models collected at baseline included, age, gender, race/ethnicity, waist circumference, smoking status, education level, chronic condition status (i.e., based on cancer, myocardial infarct, and diabetes/diabetes medication reported), weight loss attempts, energy intake and % kcal from animal protein. The final analytical sample was 46,588 after eliminating subjects with missing elements of the model. For participants with mortality recorded, follow-up time was defined as months until death date, while for other participants, follow-up was defined as months until 31 December 2015. The *p* for trend for usual dairy intake was considered significant at *p* < 0.05.

## 3. Results

### 3.1. Subject Characteristics

Table 1 provides an overview of subject characteristics at baseline and describes follow-up time before death.

### 3.2. Dairy Food Usual Intake 

Table 2 provides an overview of gender combined mean usual intakes of dairy foods by age and gender. In general, as expected, males had higher usual intakes of dairy foods compared to females.

### 3.3. Usual Dairy Food Intake and All-Cause Mortality Risk (≥19 Years Old) 

Table 3 shows the HR for usual dairy food intakes and all-cause mortality risk. Dairy food was not associated with increased risk; however, age, smoking, having a chronic condition, and male sex were associated with increased all-cause mortality risk, while higher education and trying to lose weight in the previous year were associated with decreased all-cause mortality risk. Further, being Mexican American and non-Hispanic Black (NHB) race/ethnicity was associated with increased all-cause mortality risk. 

### 3.4. Usual Dairy Food Intake and Cancer Mortality Risk, ≥19 Years Old

HR for usual dairy food intake and cancer-related mortality risk can be viewed in Table 4. No significant associations were observed for quartiles of usual dairy food intake and cancer-related mortality risk. Increased age, NHB race/ethnicity, smoking, having a chronic condition and male sex showed increased risk for cancer mortality, while trying to lose weight and having an education greater than high school were associated with reduced cancer mortality risk.

### 3.5. Usual Dairy Food Intake and CVD Mortality Risk, ≥19 Years Old 

HR for usual dairy food intake and CVD-related mortality risk can be viewed in Table 5. Usual dairy food consumption was associated with a 26% reduced risk for heart disease mortality when comparing the lowest quartile to the highest quartile [HR = 0.74; CI: 0.54–1.01; *p* = 0.05]. A 28% reduced risk in CVD mortality was seen when comparing the third quartile of usual intake with the lowest quartile of usual intake [HR = 0.72; CI: 0.58–0.90; *p* = 0.003]. Further, attainment of an education greater than high school was associated with a reduced risk, while increased age, smoking and being a male were associated with increased risk for CVD mortality risk as was waist circumference and presence of chronic conditions.

### 3.6. Usual Dairy Food Intake and CVD Mortality Risk in Older Adults (51–70 Years Old) 

HR for usual dairy food intake and CVD-related mortality risk in adults 51–70 years of age can be viewed in Table 6. Usual dairy food consumption was associated with a 31% reduced risk for CVD mortality when comparing the lowest quartile to the highest quartile [HR = 0.61; CI: 0.41–0.91; *p* = 0.015]. Results also showed a 7% risk reduction associated with 1% higher percentage of calories coming from animal protein (with a 2.5% unit change in % animal protein (% kcal), risk is estimated to be reduced by 19%). Further, attainment of an education greater than high school was associated with a reduced risk, while increased age, smoking and male sex were associated with increased risk for CVD mortality risk.

### 3.7. Usual Dairy Food Intake and CVD Mortality Risk in Older Adults (≥51 Years Old) 

HR for usual dairy food intake and CVD-related mortality risk in older adults ≥51 years of age can be viewed in Table 7. Usual dairy food intake was associated with a 31% reduced risk for heart disease mortality when comparing the lowest quartile to the highest quartile [HR = 0.69; CI: 0.54–0.89; *p* = 0.004]. A 21% reduced risk in CVD mortality was seen when comparing the third quartile of usual intake with the lowest quartile of usual intake [HR = 0.79; CI: 0.65–0.98; *p* = 0.031]. As seen with the other age groups, attainment of an education greater than high school was associated with a reduced risk, while increased age, smoking, being a male, and having a chronic condition were associated with increased risk for CVD mortality risk in adults ≥51 years old.

## 4. Discussion

Dairy foods remain an integral part of recommended US dietary patterns and are robust contributors of nutrient density to several healthy eating patterns, including the Dietary Approaches to Stopping Hypertension (DASH), Mediterranean and vegetarian diets. The present analyses demonstrate no associations between usual dairy food intake and mortality risk from all-causes, and cancer when comparing the lowest quartile to the highest quartile of consumption. Usual dairy food consumption was associated with a 26% reduced risk for heart disease mortality when comparing the lowest quartile to the highest quartile. In older adults, 51–70 and ≥51 y, usual dairy food consumption was associated with 39% and 31% reduced risk for heart disease mortality, respectively. These results contradict previous findings that have linked dairy foods to increased mortality risk. Further, our analyses using the National Cancer Institute’s usual intake method show that dairy foods, as part of a healthy dietary pattern, may help lower heart disease mortality risk.

Current and previous US dietary guidelines have recommended limiting consumption of higher fat dairy foods, particularly as such dairy foods are a source of saturated fats which are presumed to be a contributing factor to adverse cardiovascular outcomes and cardiovascular-related mortality risk. A recent review by Mozaffarian [17] established dietary priorities for cardiometabolic health included whole-fat or low-fat dairy foods and reiteration of the notion that “specific foods and overall dietary patterns, rather than single isolated nutrients, are most relevant for cardiometabolic health”. The review [17] further summarizes that longitudinal studies do not support harmful health outcomes associated with whole-fat milk consumption and chronic diseases, including cardiovascular disease, diabetes and obesity in adult populations [18,19,20,21,22]. Further, whole-fat dairy foods may offer potential positive health benefits, including reducing risks for cardiovascular and diabetes-related outcomes [23,24,25]. 

The current data indicate contradictory results for mortality risk when using the usual dairy food intake validated and preferred methodology in regression analyses in comparison to previous work [5] that have not used the method recommended by the National Cancer Institute [7], suggesting substantial methodological concerns. However, the current findings are supportive and aligned with recent data from a large 21-country cohort [26]. Similar to the current methodology, a validated food frequency questionnaire assessed dairy food intake in nearly 137,000 adults and reported that higher intakes of total dairy foods (i.e., >2 servings per day vs. no dairy food consumption) was associated with a 16% reduced risk of the composite outcome (i.e., the primary outcomes assessed the composite of mortality or major cardiovascular failure), a 17% risk reduction in total mortality, a 14% lowered risk of non-cardiovascular mortality, a 23% lowered risk of cardiovascular mortality and a 22% decreased risk of cardiovascular diseases [26]. Previous research has also examined the clinical outcomes when replacing low-fat dairy with higher-fat dairy foods. Indeed, while the original DASH diet trials [27,28] included mostly low-fat dairy foods, a subsequent clinical trial showed that the DASH diet is equally effective in contributing to cardiovascular benefits when higher-fat dairy foods are substituted for low-fat dairy foods [29]. Researchers further reduced sugars from carbohydrates in the DASH diet to compensate for additional energy provided by the higher-fat dairy foods. Results showed that the higher-fat dairy DASH diet with lowered carbohydrates had the same beneficial effect of lowering blood pressure, reducing triglycerides and very-low-density lipoprotein cholesterol levels, without elevating low-density lipoprotein cholesterol levels, which was similar to the original DASH diet with low-fat dairy foods [27,28].

Our study has limitations which are inherent in epidemiological research using NHANES data which have been previously reported [1,8,9]. The results reported here are observational in nature, and thus, only indicate an association between usual dairy food intakes and mortality outcomes. We acknowledge that there is controversy over the accuracy of the methods used to assess food intake using the methods employed in NHANES [30]. While an in-depth discussion is not warranted here as these arguments have previously been documented, we acknowledge that it has long been recognized that self-reported dietary intakes show consistent underreporting of energy intake and, to some extent, macronutrients [30]. The utilization of biomarkers may improve estimates of intake and urinary urea nitrogen could be used to correct for the underreporting of protein intake [31]. We are unaware, however, of data to suggest that reporting of usual dairy food intake would be systematically biased in those consuming higher or lower dairy food intakes, and thus would propose that our risk estimates are robust. Furthermore, mortality dates were only publicly available through 2015 [16] at the completion of the current analyses, and as such, additional and further research examining usual dairy food consumption and risk of mortality is warranted and recommended.

In conclusion, the present analyses did not demonstrate a relationship between usual dairy food intake and mortality risk from all-causes and cancer, when comparing the lowest quartile to the highest quartile of dairy food consumption. In contrast, usual dairy food intake was significantly associated with cardiovascular disease-related mortality risk. Specifically, a 26% reduced risk for heart disease mortality was observed when comparing the lowest quartile to the highest quartile of usual dairy food intake. In older adults, 51–70 and ≥51 y, usual dairy food intake was associated with 39% and 31% reduced risk for heart disease mortality, respectively. These results contradict previous findings that have linked dairy foods to increased mortality risk. The present analyses, which incorporated the National Cancer Institute’s usual intake method, provides evidence that dairy foods within a dietary pattern, may help lower heart disease mortality risk in adults.

## Figures and Tables

**Figure 1 nutrients-15-00394-f001:**
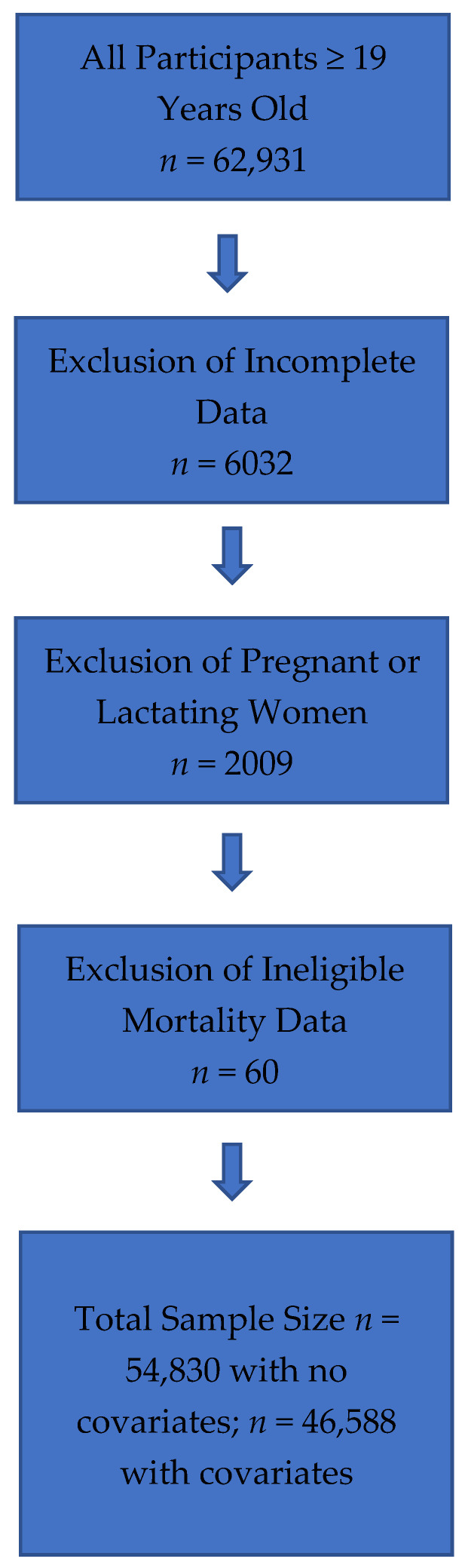
Flowchart of Total Sample and Study Exclusions for All Adults (≥19 years old) Combined.

**Table 1 nutrients-15-00394-t001:** Mean Subject Characteristics by Age Group Examined (Males and Females Combined Data).

Age (Years)	Description	Mean	SE
≥19	Age	46.50	0.20
≥19	Gender = Male (%)	49.27	0.30
≥19	Ethnicity = Mexican American (%)	7.94	0.56
≥19	Ethnicity = Other Hispanic (%)	5.08	0.50
≥19	Ethnicity = non-Hispanic White (%)	69.84	1.13
≥19	Ethnicity = non-Hispanic Black (%)	11.29	0.65
≥19	Ethnicity = Other (%)	5.84	0.28
≥19	Education ≤ High School (%)	42.96	0.77
≥19	Education = Some College (%)	31.11	0.42
≥19	Education ≥ Bachelor Degree (%)	25.84	0.69
≥19	Smoking Current (%)	21.38	0.51
≥19	Waist Circumference (cm)	97.01	0.16
≥19	Follow Up (Years)	9.35	1.10
≥19	Population *n*	206,957,440	
≥19	Sample *n*	54,830	
51–70	Age	59.37	0.08
51–70	Gender = Male (%)	47.68	0.48
51–70	Ethnicity = Mexican American (%)	4.90	0.50
51–70	Ethnicity = Other Hispanic (%)	4.04	0.47
51–70	Ethnicity = non-Hispanic White (%)	75.51	1.18
51–70	Ethnicity = non-Hispanic Black (%)	10.37	0.71
51–70	Ethnicity = Other (%)	5.19	0.38
51–70	Education ≤ High School (%)	41.98	1.00
51–70	Education = Some College (%)	29.44	0.73
51–70	Education ≥ Bachelor Degree (%)	28.54	0.94
51–70	Smoking Current (%)	19.80	0.65
51–70	Waist Circumference	101.30	0.22
51–70	Follow Up (Years)	8.61	1.38
51–70	Population *n*	60,401,257	
51–70	Sample *n*	15,771	
≥51	Age	64.20	0.11
≥51	Gender = Male (%)	46.12	0.36
≥51	Ethnicity = Mexican American (%)	4.34	0.46
≥51	Ethnicity = Other Hispanic (%)	3.74	0.47
≥51	Ethnicity = non-Hispanic White (%)	77.55	1.09
≥51	Ethnicity = non-Hispanic Black (%)	9.64	0.65
≥51	Ethnicity = Other (%)	4.73	0.32
≥51	Education ≤ High School (%)	46.19	0.88
≥51	Education = Some College (%)	27.90	0.59
≥51	Education ≥ Bachelor Degree (%)	25.82	0.80
≥51	Smoking Current (%)	16.60	0.51
≥51	Waist Circumference	100.80	0.18
≥51	Follow Up (Years)	8.10	1.18
≥51	Population *n*	82,816,918	
≥51	Sample *n*	24,697	

Data from NHANES III and NHANES 1999–2014; SE = standard error. Age signifies years of age at survey/data collection time period.

**Table 2 nutrients-15-00394-t002:** Estimated quartiles and estimated mean usual dairy intake (cup equivalents).

Age	Adults	*n*	Mean	Quartile 1	Quartile 2	Quartile 3	Quartile 4
51–70	All	15,771	1.507	<1.06	1.06–1.45	1.45–1.86	>1.86
51–70	Male	7757	1.676	<1.16	1.16–1.62	1.62–2.10	>2.10
51–70	Female	8014	1.354	<0.99	0.99–1.33	1.33–1.68	>1.68
51+	All	24,697	1.503	<1.07	1.07–1.45	1.45–1.84	>1.84
51+	Male	12,129	1.671	<1.18	1.18–1.62	1.62–2.08	>2.08
51+	Female	12,568	1.359	<1.00	1.00–1.34	1.34–1.67	>1.67
19+	All	54,830	1.638	<1.14	1.14–1.55	1.55–2.03	>2.03
19+	Male	27,200	1.869	<1.30	1.30–1.80	1.80–2.32	>2.32
19+	Female	27,630	1.414	<1.04	1.04–1.38	1.38–1.75	>1.75

Individual usual intakes were estimated using the National Cancer Institute method (Version 2.1) [7]. Data from NHANES III and NHANES 1999–2014; SE = standard error. Age signifies years of age at survey/data collection time period.

**Table 3 nutrients-15-00394-t003:** Hazard ratios for usual dairy food intake and all-cause mortality, adults ≥19 years old.

	HR	LCL	UCL	*p*
Dairy Food Usual Intake, Quartile (Q)				
Dairy Food, Q1	1.00			
Dairy Food, Q2	1.00	0.91	1.10	0.952
Dairy Food, Q3	0.98	0.90	1.08	0.722
Dairy Food, Q4	0.97	0.85	1.11	0.673
Age				
Age	1.08	1.08	1.09	<0.0001
Sex				
Female	1.00			
Male	1.61	1.47	1.81	<0.0001
Ethnicity				
Non-Hispanic White	1.00			
Mexican	1.25	1.10	1.42	0.001
Other Hispanic	0.93	0.76	1.14	0.462
Non-Hispanic Black	1.26	1.12	1.43	0.0002
Other	0.86	0.67	1.12	0.263
Lifestyle Factors and Education				
Waist Circumference	1.01	1.00	1.01	<0.0001
Non-Smoker	1.00			
Smoker	1.82	1.68	1.96	<0.0001
Chronic Condition, No	1.00			
Chronic Condition, Yes	1.70	1.59	1.82	<0.0001
Did Not Try to Lose Weight, Last Year	1.00			
Tried to Lose Weight, Last Year	0.80	0.74	0.86	<0.0001
Energy Intake (kcal)	1.00	1.00	1.00	0.001
Animal Protein (% kcal)	0.97	0.95	0.99	0.011
Education ≤ High School	1.00			
Education = Some College	0.91	0.85	0.98	0.016
Education ≥ Bachelor Degree	0.61	0.56	0.67	<0.0001

*n* = 46,588; Mortality *n* = 9708; Adults ≥19 years old; HR—Hazard Ratio; LCL—lower 95% confidence limit; and UCL—upper 95% confidence limit.

**Table 4 nutrients-15-00394-t004:** Hazard ratios for usual dairy food intake and cancer mortality in adults ≥19 years old.

	HR	LCL	UCL	*p*
Dairy Food Usual Intake, Quartile (Q)				
Dairy Food, Q1	1.00			
Dairy Food, Q2	1.07	0.89	1.28	0.470
Dairy Food, Q3	1.15	0.95	1.40	0.152
Dairy Food, Q4	0.95	0.75	1.20	0.652
Age				
Age	1.07	1.07	1.08	<0.0001
Sex				
Female	1.00			
Male	1.58	1.32	1.89	<0.0001
Ethnicity				
Non-Hispanic White	1.00			
Mexican	1.10	0.86	1.39	0.450
Other Hispanic	0.97	0.71	1.33	0.847
Non-Hispanic Black	1.34	1.09	1.65	0.006
Other	0.69	0.47	1.00	0.050
Lifestyle Factors and Education				
Waist Circumference	1.00	1.00	1.01	0.050
Non-Smoker	1.00			
Smoker	2.20	1.94	2.50	<0.0001
Chronic Condition, No	1.00			
Chronic Condition, Yes	1.78	1.52	2.09	<0.0001
Did Not Try to Lose Weight, Last Year	1.00			
Tried to Lose Weight, Last Year	0.81	0.71	0.93	0.002
Energy Intake (kcal)	1.00	1.00	1.00	0.047
Animal Protein (% kcal)	1.01	0.96	1.07	0.720
Education ≤ High School	1.00			
Education = Some College	1.02	0.88	1.20	0.795
Education ≥ Bachelor Degree	0.78	0.64	0.96	0.020

*n* = 46,588; Mortality *n* = 2025; Adults ≥19 years old; HR—Hazard Ratio; LCL—lower 95% confidence limit; and UCL—upper 95% confidence limit.

**Table 5 nutrients-15-00394-t005:** Hazard ratio for usual dairy food intake and CVD mortality risk, ≥19 years old.

	HR	LCL	UCL	*p*
Dairy Food Usual Intake, Quartile (Q)				
Dairy Food, Q1	1.00			
Dairy Food, Q2	0.91	0.74	1.12	0.370
Dairy Food, Q3	0.72	0.58	0.90	0.003
Dairy Food, Q4	0.74	0.54	1.01	0.050
Age				
Age	1.10	1.09	1.11	<0.0001
Sex				
Female	1.00			
Male	2.39	1.99	2.87	<0.0001
Ethnicity				
Non-Hispanic White	1.00			
Mexican	0.97	0.70	1.51	0.867
Other Hispanic	0.85	0.56	1.30	0.456
Non-Hispanic Black	1.18	0.94	1.47	0.147
Other	0.65	0.37	1.14	0.131
Lifestyle Factors and Education				
Waist Circumference	1.01	1.00	1.01	0.007
Non-Smoker	1.00			
Smoker	1.85	1.57	2.17	<0.0001
Chronic Condition, No	1.00			
Chronic Condition, Yes	2.02	1.71	2.38	<0.0001
Did Not Try to Lose Weight, Last Year	1.00			
Tried to Lose Weight, Last Year	0.95	0.82	1.11	0.542
Energy Intake (kcal)	1.00	1.00	1.00	0.019
Animal Protein (% kcal)	0.96	0.92	1.01	0.117
Education ≤ High School	1.00			
Education = Some College	0.98	0.82	1.16	0.799
Education ≥ Bachelor Degree	0.67	0.53	0.84	0.001

*n* = 46,588; Mortality *n* = 2025; Adults ≥19 years old; HR—Hazard Ratio; LCL—lower 95% confidence limit; and UCL—upper 95% confidence limit.

**Table 6 nutrients-15-00394-t006:** Hazard ratio for usual dairy food intake and CVD mortality risk in older adults, 51–70 years old.

	HR	LCL	UCL	*p*
Dairy Food Usual Intake, Quartile (Q)				
Dairy Food, Q1	1.00			
Dairy Food, Q2	0.73	0.51	1.03	0.071
Dairy Food, Q3	0.85	0.62	1.17	0.313
Dairy Food, Q4	0.61	0.41	0.91	0.015
Age/Sex				
Age	1.10	1.07	1.13	<0.0001
Sex				
Female	1.00			
Male	3.15	2.39	4.15	<0.0001
Ethnicity				
Non-Hispanic White	1.00			
Mexican	0.79	0.50	1.23	0.291
Other Hispanic	0.76	0.42	1.38	0.357
Non-Hispanic Black	1.17	0.83	1.65	0.373
Other	0.66	0.25	1.76	0.402
Lifestyle Factors and Education				
Waist Circumference	1.01	1.00	1.03	0.007
Non-Smoker	1.00			
Smoker	2.05	1.63	2.57	<0.0001
Chronic Condition, No	1.00			
Chronic Condition, Yes	2.50	1.86	3.37	<0.0001
Did Not Try to Lose Weight, Last Year	1.00			
Tried to Lose Weight, Last Year	0.92	0.70	1.22	0.581
Energy Intake (kcal)	1.00	1.00	1.00	0.003
Animal Protein (% kcal)	0.93	0.88	0.99	0.033
Education ≤ High School	1.00			
Education = Some College	0.79	0.59	1.05	0.105
Education ≥ Bachelor Degree	0.60	0.42	0.87	0.007

*n* = 13,680; Mortality *n* = 711; Adults ≥19 years old; HR—Hazard Ratio; LCL—lower 95% confidence limit; and UCL—upper 95% confidence limit.

**Table 7 nutrients-15-00394-t007:** Hazard ratio for usual dairy food intake and CVD mortality risk in older adults, (≥51 Years Old.

	HR	LCL	UCL	*p*
Dairy Food Usual Intake, Quartile (Q)				
Dairy Food, Q1	1.00			
Dairy Food, Q2	0.88	0.73	1.06	0.178
Dairy Food, Q3	0.79	0.65	0.98	0.031
Dairy Food, Q4	0.69	0.54	0.89	0.004
Age				
Age	1.10	1.09	1.12	<0.0001
Sex				
Female	1.00			
Male	2.37	1.99	2.83	<0.0001
Ethnicity				
Non-Hispanic White	1.00			
Mexican	0.78	0.58	1.04	0.092
Other Hispanic	0.88	0.56	1.40	0.5843
Non-Hispanic Black	1.13	0.89	1.43	0.328
Other	0.63	0.34	1.18	0.148
Lifestyle Factors and Education				
Waist Circumference	1.01	1.00	1.01	0.050
Non-Smoker	1.00			
Smoker	1.82	1.52	2.18	<0.0001
Chronic Condition, No	1.00			
Chronic Condition, Yes	1.98	1.68	2.34	<0.0001
Did Not Try to Lose Weight, Last Year	1.00			
Tried to Lose Weight, Last Year	0.97	0.83	1.13	0.661
Energy Intake (kcal)	1.00	1.00	1.00	0.002
Animal Protein (% kcal)	0.96	0.91	1.01	0.112
Education ≤ High School	1.00			
Education = Some College	0.97	0.81	1.17	0.746
Education ≥ Bachelor Degree	0.74	0.59	0.94	0.014

*n* = 21,263; Mortality *n* = 1814; Adults ≥51 years old; HR—Hazard Ratio; LCL—lower 95% confidence limit; and UCL—upper 95% confidence limit.

## Data Availability

Publicly available datasets were analyzed in this study. This data can be found here: https://wwwn.cdc.gov/nchs/nhanes (accessed on 29 December 2022).

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
