# Peer review of "Dairy Food Consumption Is Associated with Reduced Risk of Heart Disease Mortality, but Not All-Cause and Cancer Mortality in US Adults"

_nutrients, 2023, doi:10.3390/nu15020394_

Round 1

Reviewer 1 Report

This is great work since it expands research findings from many cross-sectional studies and adopts a prospective cohort design to validate/contradict previous findings. The manuscript is well written and easy to follow. Kudos to the authors. I have  only minor comments as follows.

Line 44  typo “healthy” dietary pattern

Table 1 should clarify that age is the age at baseline. 

Table 2 - please rearrange the rows so that the reference group (HR=1.0) is always at the top of each variable. Please also have two separate sections for age and for sex. Also, since the significance level is set  at 0.05, wouldn’t 95%CI (instead of 99%CI) be more appropriate to report? Same for other tables.

Table 2 - for continuous variables, does the HR reflect an increase of one unit? (e.g. a 1-year increase of age is associated with 1.08 times hazard rate) It wasn’t very clear to me and this should  be noted in the footnote. Same for other tables.

For factors including smoking status, waist circumference, chronic condition, weight  loss attempt, etc, how were changes in these variables captured during the follow-up period, or were these data collected at baseline? If the latter, please specify in the method that the models used only baseline information.

I suggest adding a supplemental table to describe the subject characteristics of this cohort. With the current draft it is difficult to visualize the distribution of age, race, waist circumference, and other subject characteristics, and follow-up time. Adding the sample size for each row in each table is also strongly recommended.

Lines 234-237 please add to specify in age groups 51+ and 51-70 y.

Author Response

Dear Reviewer,

Thank you for your feedback. Below are responses to your questions. We wish you a Happy New Year!

REVIEWER’S COMMENTS: This great work since it expands research findings from many cross-sectional studies and adopts a prospective cohort design to validate/contradict previous findings. The manuscript is well written and easy to follow. Kudos to the authors.

AUTHORS’ COMMENTS: Thank you for your feedback and I am enthusiastic to hear that the manuscript was interpreted as well written and easy to follow.

REVIEWER’S COMMENTS: Line 44 typo “healthy” dietary pattern

AUTHORS’ COMMENTS: Thank you for identifying this typo. It has been corrected.

REVIEWER’S COMMENTS: Table 1 should clarify that age is the age at baseline. 

AUTHORS’ COMMENTS: We have added a footnote to state “Age signifies years of age at survey/data collection time period.”

REVIEWER’S COMMENTS: Please rearrange the rows so that the reference group (HR=1.0) is always at the top of each variable. Please also have two separate sections for age and for sex. Also, since the significance level is set  at 0.05, wouldn’t 95%CI (instead of 99%CI) be more appropriate to report? Same for other tables.

AUTHORS’ COMMENTS: Thank you for this feedback. As recommended, we have rearranged the rows so that each reference group (HR=1.0) is always at the top of each variable. We have additionally separated age and sex as suggested. Lastly, you are correct in that the 95% CI should be reported and as such, we have revised the table to include the 95% CI values.

REVIEWER’S COMMENTS: For continuous variables, does the HR reflect an increase of one unit? (e.g. a 1-year increase of age is associated with 1.08 times hazard rate) It wasn’t very clear to me and this should be noted in the footnote. Same for other tables.

AUTHORS’ COMMENTS: Our data suggests that advancing age is linked with increased mortality risk. It does not suggest that a 1-year increase of age is associated with an 8% risk of mortality.  

REVIEWER’S COMMENTS: For factors including smoking status, waist circumference, chronic condition, weight loss attempt, etc, how were changes in these variables captured during the follow-up period, or were these data collected at baseline? If the latter, please specify in the method that the models used only baseline information.

AUTHORS’ COMMENTS: Good question. For the factors listed, these were data collected at baseline. We have added this information into the methods section.

REVIEWER’S COMMENTS: I suggest adding a supplemental table to describe the subject characteristics of this cohort. With the current draft it is difficult to visualize the distribution of age, race, waist circumference, and other subject characteristics, and follow-up time. Adding the sample size for each row in each table is also strongly recommended.

AUTHORS’ COMMENTS: An additional table has been added as suggested and is listed as Table 1. This is a great recommendation and further enriches the manuscript. Thank you for this feedback.

REVIEWER’S COMMENTS: Lines 234-237 please add to specify in age groups 51+ and 51-70 y.

AUTHORS’ COMMENTS: Please provide further clarification on this request. Thank you for your help.

Reviewer 2 Report

The manuscript “Dairy Food Consumption is Associated with Reduced Risk of Heart Disease Mortality, but Not All-Cause and Cancer Mortality in US Adults” shows a careful and convincing analyses of observational, epidemiological data, collected in the NHANES studies between 1999 to 2015. While it would, of course, be very interesting to have support for reported intake data through measurement of biomarkers, these limitations are clearly presented, and compensated to a certain extent by the large number of participants. The finding that intake of dairy, irrespective of fat content reduces hazard ratios for cardiovascular disease/mortality in older US adults is relevant.

Major points:

The selection of age ranges is not explained. While I understand the comparison of data of all participants with those of older participants (51+), it is not clear to me, what the underlying rationale was to compare the groups 51+ and 51-70. I assume, that 51-70 is fully contained within 51+, and that mortality increases above 70, explaining the mortality difference between those two groups, it would be useful for the reader to explain the choices that were made here.

Why did the authors choose overlapping groups rather than different age groups (e.g. 51-70 and >70)? If I understand Table 1 correctly, there would still be ~9000 participants in the >70 group.

 Minor points:

Line 163: Legend of Table 1: “Individual usual intakes were estimated for energy, animal protein (as % kcal) and dairy intake variables”. It is not clear to me, what I am supposed to understand from this information. I assume that: “Estimated quartiles and estimated mean usual dairy intake (cup equivalents)” is based on self-reported dairy intake in the NHANES studies, however this legend suggests, that dairy intakes were estimated from energy, animal protein and dairy intake. This is a bit confusing – and should be explained better.

The abbreviation: PA – physical activity, provided in the legends of Tables 2, 3, 4, 5 does not appear in those Tables as far as I can see, and physical activity is also not discussed elsewhere. I assume, it was present in an earlier version of the manuscript, but was later removed, without removing the entry in the legends.

Table 4 – why is there no mention of the results of quartile 3 (lowest P, highest effect)? In other Tables (e.g. Table 5), these differences are shortly mentioned. Are there any indications of there being too much of a good thing that is that a higher dairy intake may no longer reduce overall mortality?

Author Response

Dear Reviewer,

Thank you for your feedback. Below are responses to your questions. We wish you a Happy New Year!

REVIEWER’S COMMENTS: The manuscript “Dairy Food Consumption is Associated with Reduced Risk of Heart Disease Mortality, but Not All-Cause and Cancer Mortality in US Adults” shows a careful and convincing analyses of observational, epidemiological data, collected in the NHANES studies between 1999 to 2015. While it would, of course, be very interesting to have support for reported intake data through measurement of biomarkers, these limitations are clearly presented, and compensated to a certain extent by the large number of participants. The finding that intake of dairy, irrespective of fat content reduces hazard ratios for cardiovascular disease/mortality in older US adults is relevant.

AUTHORS’ COMMENTS: Thank you for your thorough review and feedback.

REVIEWER’S COMMENTS: The selection of age ranges is not explained. While I understand the comparison of data of all participants with those of older participants (51+), it is not clear to me, what the underlying rationale was to compare the groups 51+ and 51-70. I assume, that 51-70 is fully contained within 51+, and that mortality increases above 70, explaining the mortality difference between these two groups, it would be useful for the reader to explain the choices that were made here. Why did the authors choose overlapping groups rather than different age groups (e.g. 51-70 and >70)? If I understand Table 1 correctly, there would still be ~9000 participants in the >70 group.

AUTHORS’ COMMENTS: We used the 19+ age group to capture all adults, but all decided to further examine 51+ and 51-70 year-olds since previous work has identified mixed findings with animal protein intake and mortality risk in older adults. We believe that differentiating this age group is an important addition to the scientific, peer-reviewed literature.

REVIEWER’S COMMENTS: Line 163: Legend of Table 1: “Individual usual intakes were estimated for energy, animal protein (as % kcal) and dairy intake variables”. It is not clear to me, what I am supposed to understand from this information. I assume that: “Estimated quartiles and estimated mean usual dairy intake (cup equivalents)” is based on self-reported dairy intake in the NHANES studies, however this legend suggests, that dairy intakes were estimated from energy, animal protein and dairy intake. This is a bit confusing – and should be explained better.

AUTHORS’ COMMENTS:  Thank you for catching this error. The legend has been revised to read as follows: “Individual usual intakes were estimated using the National Cancer Institute method (Version 2.1)”.

REVIEWER’S COMMENTS: The abbreviation: PA – physical activity, provided in the legends of Tables 2, 3, 4, 5 does not appear in those Tables as far as I can see, and physical activity is also not discussed elsewhere. I assume, it was present in an earlier version of the manuscript, but was later removed, without removing the entry in the legends.

AUTHORS’ COMMENTS: Thank you for catching this error. The ‘PA—physical activity’ has been removed from all of the tables.

REVIEWER’S COMMENTS: Table 4 – why is there no mention of the results of quartile 3 (lowest P, highest effect)? In other Tables (e.g. Table 5), these differences are shortly mentioned. Are there any indications of there being too much of a good thing that is that a higher dairy intake may no longer reduce overall mortality?

AUTHORS’ COMMENTS: Thank you for bringing this to our attention. We have gone back and added, where results were significant, text around mortality risk reduction with other quartiles of usual intake. Overall, there are no indications from this study that higher usual dairy intake may no longer reduce overall mortality.
